# A Branch-and-Cut Approach for a Mixed Integer Linear Programming Compilation of Optimal Numeric Planning

**Ryo Kuroiwa, J. Christopher Beck**

Department of Mechanical and Industrial Engineering, University of Toronto, Toronto, Canada, ON M5S 3G8
{rkuroiwa, jcb}@mie.utoronto.ca

## Abstract

In this paper, we consider optimal numeric planning, focusing on numeric planning with simple conditions and on linear numeric planning. We propose a novel compilation of numeric planning to mixed-integer linear programming (MILP) and employ a branch-and-cut algorithm to lazily generate constraints. We empirically compare the proposed method with heuristic search algorithms and other model-based approaches including an existing MILP based method. Although the new method is not competitive with heuristic search algorithms, compared to the existing MILP based method, it finds the optimal solutions faster in some planning domains and solves two more instances in one domain.

## Introduction

Numeric planning is an extension of classical planning where states contain numeric variables. As in classical planning, heuristic search methods are used to solve numeric planning tasks. Based on the delete-relaxation in classical planning (Bonet and Geffner 2001; Hoffmann and Nebel 2001), various interval-based relaxation heuristics have been proposed for numeric planning (Hoffmann 2003; Scala et al. 2016; Aldinger and Nebel 2017; Piacentini et al. 2018b; Scala et al. 2020b). In addition, other types of heuristics used in classical planning have been adapted to numeric planning such as subgoaling based heuristics (Scala et al. 2020a), landmark heuristics (Scala et al. 2017; Kuroiwa et al. 2021), and operator-counting heuristics (Piacentini et al. 2018b; Kuroiwa et al. 2021). In terms of optimal planning, except for the max heuristics proposed by Aldinger and Nebel (2017), admissible heuristics are limited to numeric planning with simple conditions, a restricted class of numeric planning. In contrast, model-based planners that compile numeric planning tasks to optimization problems have been proposed for optimal numeric planning. Piacentini et al. (2018a) proposed a mixed-integer linear programming (MILP) based method for linear numeric planning, a superset of numeric planning with simple conditions. An optimization modulo theories (Sebastiani and Tomasi 2015) based planner can handle optimal numeric planning with state-dependent action costs in addition to linear numeric planning (Leofante et al. 2020).

In this paper, following the previous research on the integer programming (IP) model for classical planning (van den

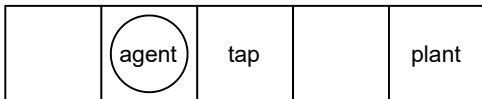

Figure 1: The example numeric planning task.

Briel, Vossen, and Kambhampati 2005), we introduce a branch-and-cut algorithm, which lazily generates constraints to the MILP model for numeric planning. We empirically compare the proposed method with the existing heuristic search algorithms and model-based planners in optimal numeric planning. Although the MILP based approaches are not competitive with the state-of-the-art heuristic search algorithms, the new MILP based method is faster and solves more instances in some numeric planning domains than the existing MILP based method.

## Motivating Example

In the existing MILP model for numeric planning (Piacentini et al. 2018a), decision variables represent which actions are applied at which time step, and the constraints required to achieve the goal conditions within a time horizon by applying actions starting from the initial state. Applying multiple actions at the same time step is allowed if they do not interfere, i.e., if they can be applied in any order and the resulting state is uniquely determined regardless of the order. The latter condition is necessary because the effects of actions can depend on values of numeric variables and are computed based on the state at the previous time step in the MILP model. We relax the former condition by allowing the application of multiple actions at the same time step if they can be applied in some order. This modification increases the number of actions that can be applied at the same time step and possibly enables the model to find an optimal solution with a shorter time horizon.

For example, consider a numeric planning task to water a plant shown in Figure 1.[1] A state consists of three numeric variables $x$, $c$, and $p$ where $x$ represents the position of an agent on a one-dimensional map, $c$ represents the amount of water carried by the agent, and $p$ represents the amount of

---

[1]This task is based on the GARDENING domain (Scala, Haslum, and Thiébaux 2016).

| action | preconditions | effects |
|---|---|---|
| move | $x \leq 3$ | $x := x + 1$ |
| move_fast | $x \leq 2$ | $x := x + 2$ |
| load | $x = 2, c \leq 4$ | $c := c + 1$ |
| pour | $x = 4, c \geq 1$ | $c := c - 1, p := p + 1$ |

Table 1: Actions in the example numeric planning task.

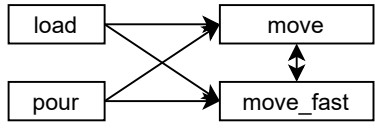

Figure 2: The action precedence relations as a graph.

water poured onto the plant. Actions move and move_fast change the position of the agent, load loads water from the tap, and pour waters the plant as shown in Table 1. In this example, move and move_fast interfere; when $x \geq 2$, applying move before move_fast is impossible since move results in $x \geq 3$, which violates the precondition of move_fast. Similarly, applying move_fast before move is also impossible when $x \geq 2$, and applying move or move_fast before load is impossible when $x = 2$. In the existing MILP model, these interfering actions cannot be applied at the same time step. However, applying move or move_fast after load is possible, and this order can be found by considering the precedence relations between actions using a graph shown in Figure 2. In our MILP model, applying interfering actions at the same time step is allowed, and the preconditions and effects of an action are computed based on the state changed by its preceding actions. Note that if actions form a cycle in the graph, applying them at the same time step is still prohibited in our model; for example, since neither applying move before move_fast or applying move_fast before move is possible, these actions cannot be applied simultaneously. Thus, the constraints that prohibit applying the actions in a cycle at the same time step are required. Since finding all cycles in advance is computationally expensive, we use a branch-and-cut algorithm which lazily generates the constraints when there are cycles in a partial solution and finds another solution satisfying the new constraints.

Suppose that $x = 1$, $c = 0$, and $p = 0$ in the initial state, and the goal condition is $p \geq 1$. The optimal plan is applying move, load, move_fast, and pour in this order. In the existing MILP model, since move and move_fast interfere with load and pour, at most one action can be applied at each time step. Therefore, the model requires the horizon of four to find the optimal plan. In contrast, in our MILP model, load can be applied before move_fast at the same time step. The optimal plan is found with the horizon of three where move is at the first time step, load and move_fast are at the second time step, and pour is at the third time step. During the search, if a found solution applies move and move_fast at the same time step, the constraints that prohibit applying the two actions at the same time step are added.

## Notation and Preliminaries

We consider a subset of numeric planning tasks defined in PDDL2.1 level 2 (Fox and Long 2003). A numeric planning task is defined as a 5-tuple $\langle \mathcal{F}_p, \mathcal{N}, \mathcal{A}, s_I, G \rangle$ where $\mathcal{F}_p$ is a set of propositions, $\mathcal{N}$ is a set of numeric variables, $\mathcal{A}$ is a set of actions, $s_I$ is the initial state, and $G$ is a set of goal conditions. A state $s$ is a tuple $\langle s_p, s_n \rangle$ where $s_p \subseteq \mathcal{F}_p$ is a set of propositions and $s_n$ is a value assignment to numeric variables. The value of $v \in \mathcal{N}$ in $s$ is denoted by $s[v]$. A numeric condition $\psi$ is represented as

$$\sum_{v \in \mathcal{N}(\psi)} w_v^{\psi} v + w_0^{\psi} \trianglerighteq 0$$

where $\mathcal{N}(\psi) \subseteq \mathcal{N}$, $w_0^{\psi} \in \mathbb{Q}$, $\forall v \in \mathcal{N}(\psi), w_v^{\psi} \in \mathbb{Q} \setminus \{0\}$, and $\trianglerighteq \in \{\geq, >\}$. A state $s$ satisfies $\psi$ if

$$\sum_{v \in \mathcal{N}(\psi)} w_v^{\psi} s[v] + w_0^{\psi} \trianglerighteq 0.$$

We denote this by $s \models \psi$. For a set of numeric conditions $\hat{\Psi}$, we say $s \models \hat{\Psi}$ if $\forall \psi \in \hat{\Psi}, s \models \psi$. $G$ is defined as a tuple $\langle G_p, G_n \rangle$ where $G_p$ is a set of propositions and $G_n$ is a set of numeric conditions. $s$ is a *goal state* if $G_p \subseteq s_p$ and $s \models G_n$.

An action $a \in \mathcal{A}$ is a triplet $\langle \mathsf{pre}(a), \mathsf{eff}(a), \mathsf{cost}(a) \rangle$ where $\mathsf{pre}(a)$ is a set of preconditions, $\mathsf{eff}(a)$ is a set of effects, and $\mathsf{cost}(a) \in \mathbb{R}_0^+$ is the cost of $a$. $\mathsf{pre}(a)$ is defined as a tuple $\langle \mathsf{pre}_p(a), \mathsf{pre}_n(a) \rangle$ where $\mathsf{pre}_p(a)$ is a set of propositions and $\mathsf{pre}_n(a)$ is a set of numeric conditions. Action $a$ is applicable in $s$ if $\mathsf{pre}_p(a) \subseteq s_p$ and $s \models \mathsf{pre}_n(a)$, and we denote this by $s \models \mathsf{pre}(a)$. By abuse of notation, we say $v \in \mathsf{pre}_n(a)$ if $\exists \psi \in \mathsf{pre}_n(a), v \in \mathcal{N}(\psi)$. $\mathsf{eff}(a)$ is defined as a triplet $\langle \mathsf{add}(a), \mathsf{del}(a), \mathsf{num}(a) \rangle$ where $\mathsf{add}(a)$ and $\mathsf{del}(a)$ are sets of propositions. $\mathsf{num}(a)$ is a set of numeric effects $v := \xi$ where $\xi$ is a linear combination of numeric variables, i.e.,

$$\xi = \sum_{u \in \mathcal{N}(\xi)} k_u^{v,a} u + k^{v,a}$$

where $N(\xi) \subseteq N$, $k^{v,a} \in \mathbb{Q}$, and $\forall u \in \mathcal{N}(\xi), k_u^{v,a} \in \mathbb{Q} \setminus \{0\}$. A concrete value of $\xi$ in $s$ is defined as follows:

$$\xi[s] = \sum_{u \in \mathcal{N}(\xi)} k_u^{v,a} s[u] + k^{v,a}.$$

Let $\mathsf{lhs}(a) = \{v \mid (v := \xi) \in \mathsf{num}(a)\}$ and $\mathsf{rhs}(a) = \{v \mid (u := \xi) \in \mathsf{num}(a), v \in \mathcal{N}(\xi)\}$. When $a$ is applied in $s$, $s$ transitions to $s[\![a]\!] = \langle s[\![a]\!]_p, s[\![a]\!]_n \rangle$ where $s[\![a]\!]_p = s_p \setminus \mathsf{del}(a) \cup \mathsf{add}(a)$ and $s[\![a]\!]_n$ is a value assignment such that $s[\![a]\!][v] = \xi[s]$ if $v \in \mathsf{lhs}(a)$ and $s[\![a]\!][v] = s[v]$ if $v \notin \mathsf{lhs}(a)$. For action $a$, we assume that $\mathsf{add}(a) \cap \mathsf{del}(a) = \emptyset$ and numeric variable $v$ is changed by at most one effect in $\mathsf{num}(a)$. An $s$-*plan* is a sequence of actions which can be sequentially applied from $s$ to make $s$ transition to a goal state. A solution for a numeric planning task is an $s_I$-plan, and an $s_I$-plan is called a *plan* for the numeric planning task. The cost of $s$-plan $\pi = \langle a_1, ..., a_n \rangle$ is defined as $\mathsf{cost}(\pi) = \mathsf{cost}(a_1) + ... + \mathsf{cost}(a_n)$. Given a numeric planning task, an

*optimal planning* problem is to find an *optimal plan*, a plan for the task which minimizes the cost.

An effect $(v := \xi)$ is called a *simple effect* if $\mathcal{N}(\xi) = \{v\}$ and $k_v^{v,a} = 1$, i.e., $\xi = v + k^{v,a}$. We denote a set of simple effects of $a$ by $\mathsf{num}_s(a)$ and a set of actions which have simple effects on $v$ by $\mathsf{se}(v)$. In contrast, we call the other effects *linear effects*. We denote a set of linear effects of $a$ by $\mathsf{num}_l(a)$ and a set of actions which have linear effects on $v$ by $\mathsf{le}(v)$. We define $\mathsf{lhs}_l(a) = \{v \mid (v := \xi) \in \mathsf{num}_l(a)\}$ and $\mathsf{rhs}_l(a) = \{v \mid (u := \xi) \in \mathsf{num}_l(a), v \in \mathcal{N}(\xi)\}$.

If $\mathcal{N} = \emptyset$ in a planning task, it is called a *classical planning* task. Particularly in the formalism used in this paper, the classical planning task is called a STRIPS task (Fikes and Nilsson 1971). If all actions have only simple effects, the task is a *numeric planning task with simple conditions* (SCT). If at least one action has linear effects, the task is a *linear numeric planning task* (LT).

A *heuristic* $h$ is a function which maps state $s$ to heuristic value $h(s) \in \mathbb{R}_0^+$. $h$ is *admissible* if $h(s)$ is less than or equal to the optimal cost of $s$-plan for all state $s$.

## Background

### State-Change IP Model for STRIPS

In the state-change IP model for STRIPS (Vossen et al. 1999), starting from the initial state, a state is changed by actions applied at multiple time steps, and decision variable $x_{a,t}$ indicates whether $a$ is applied at time step $t$. Let $s_t$ denote the state at time step $t$. $s_0 = s_I$. $a$ is applied in $s_t$ if $x_{a,t} = 1$, and $s_t$ transitions to $s_{t+1}$. The objective is to minimize the cost of the plan as follows:

$$\min \sum_{t \in \mathcal{T}} \sum_{a \in \mathcal{A}} \mathsf{cost}(a) x_{a,t} \tag{1}$$

$$\text{s.t. } x_{a,t} \in \{0,1\} \qquad \forall a \in \mathcal{A}, \forall t \in \mathcal{T} \tag{2}$$

where $\mathcal{T}$ is a set of time steps defined as $\mathcal{T} = \{0, ..., T-1\}$ and $T$ is the time horizon. Multiple actions can be applied at the same time step if the actions do not interfere, i.e., the order in which actions are applied does not affect the result. In other words, $a$ and $b$ can be applied in $s$ at the same time step if $s[\![a]\!] \models \mathsf{pre}(b)$, $s[\![b]\!] \models \mathsf{pre}(a)$, and $s[\![a]\!][\![b]\!] = s[\![b]\!][\![a]\!]$. This is ensured by the following constraint:

$$x_{a,t} + x_{b,t} \leq 1 \quad \forall a \in \mathcal{A}, \forall b \in \mathsf{interfering}(a), \forall t \in \mathcal{T} \tag{3}$$

where

$$\mathsf{interfering}(a) = \{b \in \mathcal{A} \setminus \{a\} \\ \mid \mathsf{del}(a) \cap (\mathsf{add}(b) \cup \mathsf{pre}_p(b)) \neq \emptyset\}. \tag{4}$$

There are additional decision variables representing the change of propositions.

$$y_{p,t}^a \in \{0,1\} \qquad \forall p \in \mathcal{F}_p, \forall t \in \tilde{\mathcal{T}} \tag{5}$$

$$y_{p,t}^{pa} \in \{0,1\} \qquad \forall p \in \mathcal{F}_p, \forall t \in \tilde{\mathcal{T}} \tag{6}$$

$$y_{p,t}^{pd} \in \{0,1\} \qquad \forall p \in \mathcal{F}_p, \forall t \in \tilde{\mathcal{T}} \tag{7}$$

$$y_{p,t}^m \in \{0,1\} \qquad \forall p \in \mathcal{F}_p, \forall t \in \tilde{\mathcal{T}} \tag{8}$$

where $\tilde{\mathcal{T}} = \mathcal{T} \cup \{T\}$. The constraints over these variables are defined as follows:

$$y_{p,0}^a = 1 \qquad \forall p \in s_I \tag{9}$$

$$y_{p,0}^a = 0 \qquad \forall p \in \mathcal{F}_p \setminus s_I \tag{10}$$

$$y_{p,T}^a + y_{p,T}^{pa} + y_{p,T}^m \geq 1 \qquad \forall p \in G_p \tag{11}$$

$$\sum_{a \in \mathsf{pnd}(p)} x_{a,t} \geq y_{p,t+1}^{pa} \qquad \forall p \in \mathcal{F}_p, \forall t \in \mathcal{T} \tag{12}$$

$$\sum_{a \in \mathsf{anp}(p)} x_{a,t} \geq y_{p,t+1}^a \qquad \forall p \in \mathcal{F}_p, \forall t \in \mathcal{T} \tag{13}$$

$$\sum_{a \in \mathsf{pd}(p)} x_{a,t} = y_{p,t+1}^{pd} \qquad \forall p \in \mathcal{F}_p, \forall t \in \mathcal{T} \tag{14}$$

$$x_{a,t} \leq y_{p,t+1}^{pa} \qquad \forall p \in \mathcal{F}_p, \forall a \in \mathsf{pnd}(p), \forall t \in \mathcal{T} \tag{15}$$

$$x_{a,t} \leq y_{p,t+1}^a \qquad \forall p \in \mathcal{F}_p, \forall a \in \mathsf{anp}(p), \forall t \in \mathcal{T} \tag{16}$$

$$y_{p,t}^a + y_{p,t}^m + y_{p,t}^{pd} \leq 1 \qquad \forall p \in \mathcal{F}_p, \forall t \in \tilde{\mathcal{T}} \tag{17}$$

$$y_{p,t}^{pa} + y_{p,t}^m + y_{p,t}^{pd} \leq 1 \qquad \forall p \in \mathcal{F}_p, \forall t \in \tilde{\mathcal{T}} \tag{18}$$

$$y_{p,t+1}^{pa} + y_{p,t+1}^m + y_{p,t+1}^{pd} \leq y_{p,t}^a + y_{p,t}^{pa} + y_{p,t}^m \\ \forall p \in \mathcal{F}_p, \forall t \in \mathcal{T} \tag{19}$$

where

$$\mathsf{pnd}(p) = \{a \in \mathcal{A} \mid p \in \mathsf{pre}(a), p \notin \mathsf{del}(a)\}$$
$$\mathsf{anp}(p) = \{a \in \mathcal{A} \mid p \notin \mathsf{pre}(a), p \in \mathsf{add}(a)\}$$
$$\mathsf{pd}(p) = \{a \in \mathcal{A} \mid p \in \mathsf{pre}(a), p \in \mathsf{del}(a)\}.$$

With Constraints (12) and (15), $y_{p,t}^{pa}$ indicates that $p$ was required as a precondition by actions applied at $t-1$ and remains in $s_t$. With Constraints (13) and (16), $y_{p,t}^a$ indicates that $p \in s_t$ because $p$ is added by actions applied at $t-1$. With Constraint (14), $y_{p,t}^{pd}$ indicates that $p \in s_{t-1}$ but $p \notin s_t$ because $p$ is deleted by an action applied at $t-1$. $y_{p,t}^m$ indicates that $p$ existed in $s_{t-1}$, was not required by actions applied at $t-1$, and remains in $s_t$. Constraint (19) ensures that the propositions required to be satisfied in $s_t$ are satisfied; $p$ is required in $s_t$ if either of $y_{p,t+1}^{pa}, y_{p,t+1}^m$, or $y_{p,t+1}^{pd}$ is equal to 1, and $p$ is satisfied in $s_t$ if either of $y_{p,t}^a, y_{p,t}^{pa}$, or $y_{p,t}^m$ is equal to 1. Constraints (9) and (10) specify the initial state, and Constraint (11) ensures that the goal condition is satisfied in $s_T$. Constraints (17) and (18) prevent applications of interfering actions at the same time step with Constraints (3).

### Branch-and-Cut for STRIPS

A branch-and-cut framework for classical planning (van den Briel, Vossen, and Kambhampati 2005) was originally proposed for SAS+ (Bäckström and Nebel 1995), another formalism of classical planning. Since we use STRIPS formalism, we adapt the framework to STRIPS. In the state-change

IP model, if action $a$ is applied at $t$ and $p \in \mathsf{pre}_p(a)$, action $b$ with $p \in \mathsf{del}(b)$ cannot be applied at $t$. However, in fact, we can apply $b$ after $a$ at $t$. Similarly, if $p$ is not included in $s_t$ but action $b$ with $p \in \mathsf{add}(b)$ is applied at $t$, $a$ can be applied after $b$. Such precedence relations between actions are represented by *action precedence graph* $\langle V, E \rangle$ where the set of nodes is defined as $V = \mathcal{A}$ and the set of edges is defined as

$$
\begin{aligned}
E = & \{(a, b) \mid p \in \mathcal{F}_p, a \in \mathsf{pnd}(p), b \in \mathsf{pd}(p)\} \\
& \cup \{(a, b) \mid p \in \mathcal{F}_p, a \in \mathsf{anp}(p), b \in \mathsf{pnd}(p)\}.
\end{aligned} \quad (20)
$$

If actions do not form cycles in $\langle V, E \rangle$, they can be applied at the same time step in some order. This can be ensured by the following cycle elimination constraints:

$$
\sum_{a \in V(C)} x_{a,t} \leq |V(C)| - 1 \quad (21)
$$
$$
\text{for all cycles } C \text{ in } \langle V, E \rangle, \forall t \in \mathcal{T}
$$

where $V(C)$ is the set of nodes in cycle $C$. Since the number of cycles is exponential in $|V|$, these constraints are lazily generated by a branch-and-cut algorithm. Given a solution, cycles are extracted by the following callback procedure (van den Briel, Vossen, and Kambhampati 2005):

1. Determine the subgraph $\langle V_t, E_t \rangle$ for time step $t$ consisting of all actions with $x_{a,t} > 0$.
2. For each edge $(a, b) \in E_t$, define the weight $c_{a,b} := x_{a,t} + x_{b,t} - 1$.
3. Using the Floyd-Warshall algorithm, determine the shortest path distance $d_{a,b}$ for each pair of nodes $a \in V_t$ and $b \in V_t \setminus \{a\}$ using weight $\bar{c}_{a,b} := 1 - c_{a,b}$.
4. If $d_{a,b} - c_{b,a} < 0$, there is a cycle containing $(b, a)$.

With the cycle elimination constraints, Constraint (15) is replaced with Constraint (22), which allows that precondition $p$ of action $a$ applied at $t$ is added to or deleted from $s_t$ by other actions applied at $t$.

$$
x_{a,t} \leq y_{p,t+1}^{pa} + y_{p,t+1}^{a} + y_{p,t+1}^{pd} \quad (22)
$$
$$
\forall p \in \mathcal{F}_p, \forall a \in \mathsf{pnd}(p), \forall t \in \mathcal{T}.
$$

The definition of $\mathsf{interfering}(a)$ in Equation (4) is also relaxed as follows:

$$
\mathsf{interfering}(a) = \{b \in \mathcal{A} \setminus \{a\} \mid \mathsf{del}(a) \cap \mathsf{add}(b) \neq \emptyset\}. \quad (23)
$$

This modification increases the number of pairs of actions that can be applied at the same time step and possibly reduces the number of time steps required to find a plan.

The branch-and-cut approach is similar to $\exists$-step encoding of planning as satisfiability (Wehrle and Rintanen 2007), where interfering actions can be applied at the same time step if they can be applied in some order. In $\exists$-step encoding, interfering actions are represented by the disabling-enabling graph, a directed graph that is similar to the action precedence graph but considers only interference in reachable states. While the branch-and-cut approach prohibits the simultaneous application of actions that form a cycle in the action precedence graph, $\exists$-step encoding allows it by imposing a fixed ordering on actions in all strongly connected components of the disabling-enabling graph. Using a fixed ordering in an IP model is an interesting topic for future work.

## MILP Model for Numeric Planning

Recent research has extended the state-change IP model for STRIPS to numeric planning (Piacentini et al. 2018a). The model has additional decision variable $y_t^v$, which represents the value of $v \in \mathcal{N}$ in $s_t$.

$$
y_t^v \in \mathbb{Q} \qquad \forall v \in \mathcal{N}, \forall t \in \tilde{\mathcal{T}}. \quad (24)
$$

For the numeric variables in the initial state,

$$
y_0^v = s_I[v] \qquad \forall v \in \mathcal{N}. \quad (25)
$$

For the numeric goal conditions,

$$
\sum_{v \in \mathcal{N}} w_v^{\psi} y_T^v + w_0^{\psi} \unrhd 0 \qquad \forall \psi \in G_n. \quad (26)
$$

For the numeric preconditions of actions,

$$
\sum_{v \in \mathcal{N}(\psi)} w_v^{\psi} y_t^v + w_0^{\psi} \unrhd m_{\psi,t}(1 - x_{a,t}) \quad (27)
$$
$$
\forall a \in \mathcal{A}, \forall \psi \in \mathsf{pre}_n(a), \forall t \in \mathcal{T}
$$

where $m_{\psi,t}$ is the lower bound of the left-hand side computed by Equation (34). For the numeric effects,

$$
y_{t+1}^v \leq y_t^v + \sum_{a \in \mathsf{se}(v)} k^{v,a} x_{a,t} + M_{v,t+1}^{\mathsf{step}} \sum_{a \in \mathsf{le}(v)} x_{a,t} \quad (28)
$$
$$
\forall v \in \mathcal{N}, \forall t \in \mathcal{T}
$$
$$
y_{t+1}^v \geq y_t^v + \sum_{a \in \mathsf{se}(v)} k^{v,a} x_{a,t} + m_{v,t+1}^{\mathsf{step}} \sum_{a \in \mathsf{le}(v)} x_{a,t} \quad (29)
$$
$$
\forall v \in \mathcal{N}, \forall t \in \mathcal{T}
$$
$$
y_{t+1}^v \leq k^{v,a} + \sum_{u \in \mathcal{N}(\xi)} k_u^{v,a} y_t^u + M_{v,t+1}^a (1 - x_{a,t}) \quad (30)
$$
$$
\forall a \in \mathcal{A}, \forall (v := \xi) \in \mathsf{num}_l(a), \forall t \in \mathcal{T}
$$
$$
y_{t+1}^v \geq k^{v,a} + \sum_{u \in \mathcal{N}(\xi)} k_u^{v,a} y_t^u + m_{v,t+1}^a (1 - x_{a,t}) \quad (31)
$$
$$
\forall a \in \mathcal{A}, \forall (v := \xi) \in \mathsf{num}_l(a), \forall t \in \mathcal{T}
$$

where $M_{v,t+1}^{\mathsf{step}}$ and $m_{v,t+1}^{\mathsf{step}}$ are upper and lower bounds of $y_{t+1}^v - y_t^v$ computed by Equations (35) and (36), and $M_{v,t+1}^a$ and $m_{v,t+1}^a$ are upper and lower bounds of $y_{t+1}^v - k^{v,a} - \sum_{u \in \mathcal{N}(\xi)} k_u^{v,a} y_t^u$ computed by Equations (40) and (41). Constraints (28) and (29) correspond to simple effects, and Constraints (30) and (31) correspond to linear effects. Considering numeric preconditions and effects, Equation (4) is modified as follows:

$$
\begin{aligned}
& \mathsf{interfering}(a) \\
& = \{b \in \mathcal{A} \setminus \{a\} \mid \mathsf{del}(a) \cap (\mathsf{add}(b) \cup \mathsf{pre}_p(b)) \neq \emptyset\} \\
& \cup \{b \in \mathcal{A} \setminus \{a\} \mid \exists v \in \mathsf{lhs}(a), v \in \mathsf{pre}_n(b)\} \\
& \cup \{b \in \mathcal{A} \setminus \{a\} \mid \mathsf{lhs}(a) \cap \mathsf{rhs}_l(b) \neq \emptyset\}.
\end{aligned} \quad (32)
$$

Equation (32) is used in the original paper (Piacentini et al. 2018a) but is unnecessarily strict; when $b$ is applicable in $s$, if $a$ has only simple effects on $\forall v \in \mathsf{pre}_n(b)$ and the net

effects of $a$ on $\forall \psi \in \mathsf{pre}_n(b)$ are positive, $b$ is also applicable in $s[\![a]\!]$. Using this fact, we use the following equation instead of Equation (32).

$$
\begin{aligned}
&\mathsf{interfering}(a) \\
&= \{b \in \mathcal{A} \setminus \{a\} \mid \mathsf{del}(a) \cap (\mathsf{add}(b) \cup \mathsf{pre}_p(b)) \neq \emptyset\} \\
&\cup \{b \in \mathcal{A} \setminus \{a\} \mid \exists \psi \in \mathsf{pre}_n(b), \mathsf{net}(a, \psi) < 0\} \\
&\cup \{b \in \mathcal{A} \setminus \{a\} \mid \mathsf{lhs}(a) \cap \mathsf{rhs}_l(b) \neq \emptyset\}
\end{aligned}
\quad (33)
$$

where $\mathsf{net}(a, \psi) = \sum_{v \in \mathcal{N}(\psi)} w_v^\psi k_a^v + w_0^\psi$ if $a$ has only simple effects on $\mathcal{N}(\psi)$, $\mathsf{net}(a, \psi) = 0$ if $a$ has no effect on $\mathcal{N}(\psi)$, and $\mathsf{net}(a, \psi) = -\infty$ otherwise since linear effects change values depending on the state.

**Big-M Values** Let $M_{v,t}$ and $m_{v,t}$ be upper and lower bounds of $y_t^v$. The big-M value in Constraint (27), $m_{\psi,t}$, is computed as follows:

$$
m_{\psi,t} = \sum_{v \in \mathcal{N}(\psi)^-} w_v^\psi M_{v,t} + \sum_{v \in \mathcal{N}(\psi)^+} w_v^\psi m_{v,t} + w_0^\psi \quad (34)
$$

where $\mathcal{N}(\psi)^+ = \{v \in \mathcal{N}(\psi) \mid w_v^\psi > 0\}$ and $\mathcal{N}(\psi)^- = \{v \in \mathcal{N}(\psi) \mid w_v^\psi < 0\}$. $M_{v,t}^{\mathrm{step}}$ and $m_{v,t}^{\mathrm{step}}$ in Constraints (28) and (29) are computed as follows:

$$
M_{v,t}^{\mathrm{step}} = M_{v,t} - m_{v,t-1} \quad (35)
$$
$$
m_{v,t}^{\mathrm{step}} = m_{v,t} - M_{v,t-1}. \quad (36)
$$

Finally, $M_{v,t}^a$ and $m_{v,t}^a$ in Constraints (30) and (31) are computed as follows:

$$
\begin{aligned}
M_{v,t}^a = M_{v,t} - k^{v,a} &- \sum_{u \in \mathcal{N}(\xi)^-} k_u^{v,a} M_{u,t-1} \\
&- \sum_{u \in \mathcal{N}(\xi)^+} k_u^{v,a} m_{u,t-1}
\end{aligned}
\quad (37)
$$

$$
\begin{aligned}
m_{v,t}^a = m_{v,t} - k^{v,a} &- \sum_{u \in \mathcal{N}(\xi)^+} k_u^{v,a} M_{u,t-1} \\
&- \sum_{u \in \mathcal{N}(\xi)^-} k_u^{v,a} m_{u,t-1}
\end{aligned}
\quad (38)
$$

where $\mathcal{N}(\xi)^+ = \{u \in \mathcal{N}(\xi) \mid k_u^{v,a} > 0\}$ and $\mathcal{N}(\xi)^- = \{u \in \mathcal{N}(\xi) \mid k_u^{v,a} < 0\}$. $M_{v,t}$ and $m_{v,t}$ are computed

through the following equations:

$$
M_{v,0} = m_{v,0} = s_I[v] \quad (39)
$$

$$
M_{v,t} = \max \left\{ M_{v,t-1} + \sum_{a \in \mathsf{se}^+(v)} k^{v,a}, \ \max_{a \in \mathsf{le}(v)} \bar{a}_l(v,t) \right\}
\quad (40)
$$

$$
m_{v,t} = \min \left\{ m_{v,t-1} + \sum_{a \in \mathsf{se}^-(v)} k^{v,a}, \ \min_{a \in \mathsf{le}(v)} \underline{a}_l(v,t) \right\}
\quad (41)
$$

$$
\bar{a}_l(v,t) = \sum_{u \in \mathcal{N}(\xi)^+} k_u^{v,a} M_{u,t-1} + \sum_{u \in \mathcal{N}(\xi)^-} k_u^{v,a} m_{u,t-1}
\quad (42)
$$

$$
\underline{a}_l(v,t) = \sum_{u \in \mathcal{N}(\xi)^+} k_u^{v,a} m_{u,t-1} + \sum_{u \in \mathcal{N}(\xi)^-} k_u^{v,a} M_{u,t-1}
\quad (43)
$$

where $\mathsf{se}^+(v) = \{a \in \mathsf{se}(v) \mid k^{v,a} > 0\}$ and $\mathsf{se}^-(v) = \{a \in \mathsf{se}(v) \mid k^{v,a} < 0\}$.

**Landmark Constraints** A fact landmark is a proposition that is added by all plans, and an action landmark is an action that is applied by all plans (Hoffmann, Porteous, and Sebastia 2004). Let $F_L$ be a set of fact landmarks and $A_L$ be a set of action landmarks. In the MILP compilation, the following valid inequalities are used (Piacentini et al. 2018a).

$$
\sum_{t \in \tilde{\mathcal{T}}} y_{p,t}^a + y_{p,t}^{pa} + y_{p,t}^m \geq 1 \qquad \forall p \in F_L \quad (44)
$$

$$
\sum_{t \in \mathcal{T}} x_{a,t} \geq 1 \qquad \forall a \in A_L. \quad (45)
$$

We extract fact and action landmarks in SCTs using the algorithm proposed by Scala et al. (2017). In LTs, we do not use these constraints.

**Relevance Constraints** In STRIPS, action $a$ is relevant if $\exists p \in \mathsf{add}(a), p \in G_p$ or $\exists p \in \mathsf{pre}_p(b)$ where $b$ is a relevant action (Imai and Fukunaga 2015). In numeric planning, in addition to the original condition, $a$ is relevant if $\exists \psi \in G_n : \mathsf{net}(a, \psi) > 0$, $\exists \psi \in \mathsf{pre}_n(b) : \mathsf{net}(a, \psi) > 0$, $\exists v \in \mathsf{pre}_n(b) : a \in \mathsf{le}(v)$, or $\exists v \in \mathsf{rhs}_l(b) : a \in \mathsf{se}(v) \cup \mathsf{le}(v)$ where $b$ is a relevant action (Piacentini et al. 2018b). The following valid equations are used in the MILP compilation (Piacentini et al. 2018a).

$$
x_{a,t} = 0 \quad \forall a \in \mathcal{A} \text{ s.t. } a \text{ is not relevant}, \forall t \in \mathcal{T}. \quad (46)
$$

## Iterative Time Horizon Allocation

To find an optimal plan, the iterative time horizon allocation method is proposed (Piacentini et al. 2018a). Suppose that plan $\pi_T^*$ is found with horizon $T$. If $\mathsf{cost}(\pi_T^*) = l$ where $l$ is a known lower bound of the optimal cost, $\pi_T^*$ is optimal. Given admissible heuristic $h$, $h(s_I)$ can be used as the lower bound. If $\mathsf{cost}(\pi_T^*) \leq T \cdot \min_{a \in \mathcal{A}} \mathsf{cost}(a)$, $\pi_T^*$ is optimal because the right-hand side is the lower bound of the cost of plans with horizon $T' \geq T$. Otherwise, when

$\min_{a\in\mathcal{A}} \mathsf{cost}(a) > 0$, solving the MILP model with horizon $\hat{T} = \mathsf{cost}(\pi_T^*)/\min_{a\in\mathcal{A}} \mathsf{cost}(a)$ gives us an optimal plan since $\hat{T}$ is the upper bound of time steps required by an optimal plan. When solving the MILP model with $\hat{T}$, the following additional constraints are added:

$$\sum_{a\in\mathcal{A}} x_{a,t} \leq 1 \qquad \forall t \in \mathcal{T} \qquad (47)$$

$$\sum_{a\in\mathcal{A}} x_{a,t+1} \leq \sum_{a\in\mathcal{A}} x_{a,t} \qquad \forall t \in \mathcal{T}\setminus\{T-1\}. \qquad (48)$$

The first constraint ensures that at most one action is applied at each time step, and the second constraint breaks the symmetries caused by time steps where no action is applied. Note that this method is not applicable in planning tasks with $\min_{a\in\mathcal{A}} \mathsf{cost}(a) = 0$. The MILP based method starts from some $T$, increments $T$ by one if the model is infeasible, uses $\hat{T}$ if a plan is found but its optimality is not guaranteed, and terminates when the optimal plan is found.

## Branch-and-Cut for Numeric Planning

Following the branch-and-cut framework for classical planning, we introduce precedence relations between actions using graph $\langle V, E\rangle$. Suppose that $\langle V, E\rangle$ is given. Even when $a$ is not applicable in $s_t$, it may be applicable at $t$ if $b$ is applied at $t$ such that $(b,a) \in E$ and $\exists\psi \in \mathsf{pre}_n(a), \mathsf{net}(b,\psi) > 0$. However, even when $a$ is applicable in $s_t$, it may not be applicable at $t$ if $b$ is applied at $t$ such that $(b,a) \in E$ and $\exists\psi \in \mathsf{pre}_n(a), \mathsf{net}(b,\psi) < 0$. Therefore, Constraint (27) is modified as follows:

$$\sum_{v\in\mathcal{N}(\psi)} w_v^\psi y_t^v + w_0^\psi + \sum_{(b,a)\in E} \mathsf{net}(b,\psi)x_{b,t}$$
$$\trianglerighteq (m_{\psi,t} + \sum_{(b,a)\in E:\mathsf{net}(b,\psi)<0} \mathsf{net}(b,\psi))(1-x_{a,t}) \qquad (49)$$
$$\forall\psi \in \mathsf{pre}_n(a), \forall t \in \mathcal{T}.$$

The left-hand side is now considering effects of $b$, and the big-M value in the right-hand side is modified accordingly. Similarly, the linear effects of $a$ at time step $t$ must consider the change of values of $\forall v \in \mathsf{rhs}_l(a)$ within $t$ caused by simple effects of actions $b$ with $(b,a) \in E$. Constraints (30) and (31) are modified as follows:

$$y_{t+1}^v \leq k^{v,a} + \sum_{u\in\mathcal{N}(\xi)} k_u^{v,a}(y_t^u + \sum_{b\in\mathsf{se}(u):(b,a)\in E} k^{u,a}x_{b,t})$$
$$+ M_{v,t+1}^a(1-x_{a,t})$$
$$\forall a \in \mathcal{A}, \forall(v := \xi) \in \mathsf{num}_l(a), \forall t \in \mathcal{T} \qquad (50)$$

$$y_{t+1}^v \geq k^{v,a} + \sum_{u\in\mathcal{N}(\xi)} k_u^{v,a}(y_t^u + \sum_{b\in\mathsf{se}(u):(b,a)\in E} k^{u,a}x_{b,t})$$
$$+ m_{v,t+1}^a(1-x_{a,t})$$
$$\forall a \in \mathcal{A}, \forall(v := \xi) \in \mathsf{num}_l(a), \forall t \in \mathcal{T} \qquad (51)$$

where big-M values are also modified as

$$M_{v,t}^a = M_{v,t} - k^{v,a}$$
$$- \sum_{u\in\mathcal{N}(\xi)^-} k_u^{v,a}(M_{u,t-1} + \sum_{b\in\mathsf{se}^+(u):(b,a)\in E} k^{u,b})$$
$$- \sum_{u\in\mathcal{N}(\xi)^+} k_u^{v,a}(m_{u,t-1} + \sum_{b\in\mathsf{se}^-(u):(b,a)\in E} k^{u,b})$$
$$(52)$$

$$m_{v,t}^a = m_{v,t} - k^{v,a}$$
$$- \sum_{u\in\mathcal{N}(\xi)^+} k_u^{v,a}(M_{u,t-1} + \sum_{b\in\mathsf{se}^+(u):(b,a)\in E} k^{u,b})$$
$$- \sum_{u\in\mathcal{N}(\xi)^-} k_u^{v,a}(m_{u,t-1} + \sum_{b\in\mathsf{se}^-(u):(b,a)} k^{u,b})$$
$$(53)$$

In addition, Equations (42) and (43) are modified as follows:

$$\overline{a}_l(v,t) = \sum_{u\in\mathcal{N}(\xi)^+} k_u^{v,a}(M_{u,t-1} + \sum_{b\in\mathsf{se}^+(u):(b,a)\in E} k^{u,b})$$
$$+ \sum_{u\in\mathcal{N}(\xi)^-} k_u^{v,a}(m_{u,t-1} + \sum_{b\in\mathsf{se}^-(u):(b,a)\in E} k^{u,b})$$
$$(54)$$

$$\underline{a}_l(v,t) = \sum_{u\in\mathcal{N}(\xi)^+} k_u^{v,a}(m_{u,t-1} + \sum_{b\in\mathsf{se}^-(u):(b,a)\in E} k^{u,b})$$
$$+ \sum_{u\in\mathcal{N}(\xi)^-} k_u^{v,a}(M_{u,t-1} + \sum_{b\in\mathsf{se}^+(u):(b,a)\in E} k^{u,b}).$$
$$(55)$$

Now we define the actual precedence relations. If $b$ is applied before $a$ and $b$ has linear effects on the preconditions of $a$, whether $a$ is applicable or not cannot be represented by Constraint (49) because the linear effects depend on the state in which $b$ is applied. Therefore, such a $b$ must be applied after $a$. If $a$ is applied before $b$ and $a$ has simple effects on variables in $\mathsf{rhs}_l(b) \cup \mathsf{lhs}_l(b)$, the value of $v \in \mathsf{lhs}_l(b)$ at $t+1$ is represented by $y_{t+1}^v$ in Constraints (50) and (51); if $a$ changes variables in $\mathsf{rhs}_l(b)$, the change is considered in the constraints. If $a$ changes $v$, the value of $v$ is just overwritten by $b$. In contrast, if such an $a$ is applied after $b$, $y_{t+1}^v$ may contradict the actual value of $v$ at $t+1$. Therefore, such an $a$ must be applied before $b$. Based on the above observations, we first define a subset of precedence relations $E' \subseteq E$.

$$E' = \{(a,b) \mid p \in \mathcal{F}_p, a \in \mathsf{pnd}(p), b \in \mathsf{pd}(p)\}$$
$$\cup \{(a,b) \mid p \in \mathcal{F}_p, a \in \mathsf{anp}(p), b \in \mathsf{pnd}(p)\}$$
$$\cup \{(a,b) \mid a \in \mathcal{A}, v \in \mathsf{pre}_n(a), b \in \mathsf{le}(v)\setminus\{a\}\}$$
$$\cup \{(a,b) \mid v \in \mathcal{N}, a \in \mathsf{se}(v), b \in \mathcal{A}\setminus\{a\}, v \in \mathsf{lhs}_l(b)\}$$
$$\cup \{(a,b) \mid v \in \mathcal{N}, a \in \mathsf{se}(v), b \in \mathcal{A}\setminus\{a\}, v \in \mathsf{rhs}_l(b)\}.$$
$$(56)$$

Suppose that $(a,b) \notin E'$ and $(b,a) \notin E'$ for $a, b \in \mathcal{A}$. When $\mathsf{net}(b,\psi) \geq 0$ for all $\psi \in \mathsf{pre}_n(a)$ and $s \models \mathsf{pre}(a)$, applying $a$ after $b$ is possible since $s[\![b]\!]$ always satisfies $\forall\psi \in \mathsf{pre}_n(a)$. Conversely, when $\mathsf{net}(b,\psi) < 0$ for some

| | $C^{\text{SC}}$ | $C^{\text{SC}}_{\text{cut}}$ |
|---|---|---|
| Objective | (1) | (1) |
| Variables | (2), (5)-(8), (24) | (2), (5)-(8), (24) |
| $s_I$ | (9), (10), (25) | (9), (10), (25) |
| $G$ | (11), (26) | (11), (26) |
| pre | (12), (15)*, (27)* | (12), (22)*, (49)* |
| eff | (13), (14), (16)-(19), | (13),(14), (16)-(19) |
| | (28), (29), (30)*, (31)* | (28), (29), (50)*, (51)* |
| interfering | (3), (33)* | (3), (58)* |
| Big-M | (34)–(36), (37)*, (38)* | (34)–(36), (52)*, (53)*, |
| | (39)-(41), (42)*, (43)* | (39)-(41), (54)*, (55)* |
| Valid | (44)-(46) | (44)-(46) |
| $T$ | (47), (48) | (47), (48) |
| $E$ | - | (21)*, (57)* |

Table 2: Constraints used in the MILP based methods. Constraints not included in the other model are marked with '*'.

$\psi \in \text{pre}_n(a)$ and $s \models \text{pre}(a)$, applying $a$ after $b$ is impossible if $s[\![b]\!]$ does not satisfy $\psi$. Therefore, when $\text{net}(b, \psi) < 0$, we add edge $(a, b)$ to $E$ so that $b$ is applied after $a$.

$$E = E' \cup \{(a, b) \mid a, b \in \mathcal{A}, \psi \in \text{pre}_n(a), \text{net}(b, \psi) < 0,$$
$$(a, b) \notin E', (b, a) \notin E', a \neq b\} \tag{57}$$

Now, Equation (33) is relaxed as follows:

$$\text{interfering}(a) = \{b \in \mathcal{A} \setminus \{a\} \mid \text{del}(a) \cap \text{add}(b) \neq \emptyset\}$$
$$\cup \{b \in \mathcal{A} \setminus \{a\} \mid \text{lhs}_l(a) \cap \text{rhs}_l(b) \neq \emptyset\}. \tag{58}$$

### Preprocessing Action Precedence

In the branch-and-cut algorithm, each time a solution is found, the callback procedure to find cycles in graph $\langle V, E \rangle$ is executed. However, if the graph has no cycle, we can skip the callback procedure and just use a branch-and-bound algorithm. We can check if the graph has cycles using topological sorting. We adopt the following procedure before solving the task:

1. $\forall (a, b) \in E$, if $(b, a) \in E$, remove $(a, b)$ and $(b, a)$ from $E$ and update $\text{interfering}(a)$ to $\text{interfering}(a) \cup \{b\}$ and $\text{interfering}(b)$ to $\text{interfering}(b) \cup \{a\}$.

2. Execute topological sorting of $\langle V, E \rangle$. If there is no cycle, disable the callback procedure.

In this procedure, we remove cycles formed by only two actions to increase the chance that there is no cycle.

### Summary of the MILP Based Methods

We denote the baseline MILP based method proposed by Piacentini et al. (2018a) by $C^{\text{SC}}$ and our MILP based method using the branch-and-cut algorithm by $C^{\text{SC}}_{\text{cut}}$. The constraints used in the MILP based methods are summarized in Table 2.

## Experimental Evaluation

### Configurations of the MILP Based Methods

Both methods use the iterative time horizon allocation method, and the delete relaxation heuristic $h^C_{\text{IP}}$ (Piacentini

et al. 2018b) is used to compute a lower bound of the cost of an optimal plan. $h^C_{\text{IP}}$ ignores numeric conditions in LTs since they only support SCTs. Since computing $h^C_{\text{IP}}(s_I)$ generates a relaxed plan, the length of the relaxed plan is used as the initial time horizon. These are the same settings used in Piacentini et al. (2018a). If the length of the relaxed plan is zero, we use $T = 1$ as the initial time horizon. In $C^{\text{SC}}_{\text{cut}}$, the callback procedure is executed at every node in the branch-and-bound tree. If one cycle is found at some time step, cycle elimination constraints are added for all time steps, and the callback procedure is terminated. Therefore, at most one cycle is found at each execution of the callback procedure.

### Experimental Settings

We implemented the MILP based methods on top of Numeric Fast Downward (NFD) (Aldinger and Nebel 2017)[2] using C++11 with GCC 7.5.0 and Gurobi 9.1.1 on Ubuntu 18.04. However, for the heuristics, we use CPLEX 12.10 since Gurobi is not supported by the implementations of the heuristics in NFD. All experiments are run on an Intel(R) Xeon(R) CPU E5-2620 @2.00GHz processor. We use a 30 minutes time limit and 4GB memory limit for each instance.

Numeric planning domains in SCTs and LTs are taken from the literature (Scala, Haslum, and Thiébaux 2016; Scala et al. 2017, 2020a; Li et al. 2018) and IPC 3 removing duplicate instances. The instances in SCTs are basically the same as those used by Kuroiwa et al. (2021), but we use the unit cost versions of DEPOTS, ROVERS, and SATELLITE since they have actions with the costs of zero, which are not supported by the iterative time horizon allocation method. We also exclude FARMLAND-SAT since NFD runs out of memory when preprocessing the PDDL files.

### Comparison of the MILP Based Methods

We show the comparison of the two MILP based methods in Table 3. 'C' is coverage, the number of solved instances within the limits, 'Time' is the time to find the optimal plan, 'I' is the number of iterations used by the iterative time horizon allocation method, and 'Constraints' is the number of constraints in the model used in the last iteration. 'Cycles' is the number of eliminated cycles in $\langle V, E \rangle$ by the branch-and-cut algorithm, and '-' indicates that there are no cycles in the domain. In SCTs, only three domains have cycles in $\langle V, E \rangle$. In LTs, there are no cycles at all. $C^{\text{SC}}_{\text{cut}}$ has fewer constraints than $C^{\text{SC}}$ in the majority of domains. In 7 domains, $C^{\text{SC}}_{\text{cut}}$ reduces the number of iterations compared to $C^{\text{SC}}$, which means that $C^{\text{SC}}_{\text{cut}}$ finds a feasible solution with fewer time steps. Among these domains, $C^{\text{SC}}_{\text{cut}}$ achieves speed up in GARDENING-SAT, FO-COUNTERS-INV, and FO-COUNTERS-RND and solves two more instance in FO-COUNTERS-RND.

### Comparison with Other Approaches

Next, we compare the MILP based methods with other approaches. In SCTs, we evaluate A* search (Hart, Nilsson,

---

[2]https://github.com/Kurorororo/numeric-fast-downward

| | $C^{\text{SC}}$ | | | | $C^{\text{SC}}_{\text{cut}}$ | | | | Cycles |
|---|---|---|---|---|---|---|---|---|---|
| | C | T | I | Constraints | C | T | I | Constraints | |
| SMALLCOUNTERS (8) | 8 | 0.07 | 1.00 | 6721.00 | 8 | 0.07 | 1.00 | 6721.00 | - |
| COUNTERS (8) | 8 | 373.24 | 1.00 | 681677.00 | 8 | **346.42** | 1.00 | 681677.00 | - |
| COUNTERS-INV (11) | 11 | 19.81 | 1.00 | 769539.36 | 11 | **19.12** | 1.00 | 769539.36 | - |
| COUNTERS-RND (33) | 33 | 13.13 | 1.00 | 516883.67 | 33 | **12.94** | 1.00 | 516883.67 | - |
| FARMLAND (30) | 30 | **3.46** | 1.00 | 136138.67 | 30 | 3.50 | 1.00 | 136138.67 | - |
| GARDENING (63) | 63 | **60.43** | 2.60 | 15596.08 | 63 | 64.06 | **2.08** | **14871.95** | - |
| GARDENING-SAT (51) | 12 | 393.25 | 2.50 | 33032.58 | 12 | **350.91** | **1.92** | **31137.00** | - |
| SAILING (40) | 38 | 82.52 | 1.00 | 94052.08 | 38 | **26.15** | 1.00 | **87346.08** | - |
| SAILING-SAT (40) | 8 | **8.23** | 1.00 | 64931.38 | 8 | 9.33 | 1.00 | **60346.38** | - |
| DEPOTS (20) | 2 | **128.87** | 1.50 | 56808.00 | 2 | 165.85 | 1.50 | **32523.00** | 0.00 |
| ROVERS (20) | **7** | 209.28 | 1.17 | 55902.17 | 6 | **136.77** | 1.17 | **48910.83** | 6.67 |
| SATELLITE (20) | 3 | **196.74** | 3.67 | 44481.33 | 3 | 592.31 | **2.00** | **41287.33** | 32.33 |
| TOTAL (344) | **223** | - | - | - | 222 | - | - | - | - |
| FO-COUNTERS (20) | 3 | **2.37** | 3.67 | 2287.67 | 3 | 3.15 | **2.67** | **2215.33** | - |
| FO-COUNTERS-INV (20) | 2 | 0.93 | 4.50 | 2108.00 | 2 | **0.38** | **3.00** | **1933.00** | - |
| FO-COUNTERS-RND (60) | 11 | 2.03 | 4.36 | 4140.64 | 13 | **1.90** | **2.91** | 4067.18 | - |
| FO-FARMLAND (50) | 4 | **247.06** | 13.00 | **2245.00** | 4 | 271.49 | **9.50** | 2484.75 | - |
| FO-SAILING (20) | **1** | - | - | - | 0 | - | - | - | - |
| TOTAL (170) | 21 | - | - | - | **22** | - | - | - | - |

Table 3: Comparison of the MILP models. Coverage ('C'), the time ('T') in seconds, the number of iterations ('I'), and the number of constraints ('Constraints') are shown. 'T', 'I', and 'Constraints' are averaged over instances solved by the both methods. 'Cycles' is the number of eliminated cycles and averaged over instances solved by $C^{\text{SC}}_{\text{cut}}$. '-' in 'Cycles' indicates that the domain has no cycle.

| | $h^C_{\text{IP}}$ | | $h^{\text{LM-cut},SEQ}_{\text{LP}}$ | | $C^{\text{SC}}$ | | $C^{\text{SC}}_{\text{cut}}$ | |
|---|---|---|---|---|---|---|---|---|
| | C | T | C | T | C | T | C | T |
| SMALLCOUNTERS (8) | 8 | 1.56 | 8 | **0.04** | 8 | 0.07 | 8 | 0.07 |
| COUNTERS (8) | 4 | 610.01 | **8** | **2.78** | **8** | 15.74 | **8** | 16.02 |
| COUNTERS-INV (11) | 6 | 250.83 | **11** | **1.56** | **11** | 3.16 | **11** | 3.05 |
| COUNTERS-RND (33) | 21 | 360.62 | **33** | **1.91** | **33** | 3.05 | **33** | 3.02 |
| FARMLAND (30) | 30 | 43.01 | 30 | **1.50** | 30 | 3.46 | 30 | 3.50 |
| GARDENING (63) | 63 | 12.26 | 63 | **0.31** | 63 | 60.43 | 63 | 64.06 |
| GARDENING-SAT (51) | 14 | 51.05 | **15** | **1.05** | 12 | 393.25 | 12 | 350.91 |
| SAILING (40) | **40** | 41.52 | **40** | **0.58** | 38 | 82.52 | 38 | 26.15 |
| SAILING-SAT (40) | **24** | 33.17 | 12 | **0.67** | 8 | 8.23 | 8 | 9.33 |
| DEPOTS (20) | 2 | 47.28 | **7** | **0.15** | 2 | 128.87 | 2 | 165.85 |
| ROVERS (20) | **8** | 261.24 | 7 | **130.03** | 7 | 209.28 | 6 | 136.77 |
| SATELLITE (20) | 3 | 8.21 | 3 | **1.43** | 3 | 287.75 | 3 | 857.27 |
| TOTAL (344) | 223 | - | **237** | - | 223 | - | 222 | - |

Table 4: Results on SCTs. Coverage ('C') and the time ('T') in seconds are shown. 'T' is averaged over instances solved by all methods.

| | $h^{\text{blind}}$ | | $h^C_{\text{IP}}$ | | $h^{\text{iimax}}$ | | $h^{\text{irmax}}$ | | OMTPlan | | $C^{\text{SC}}$ | | $C^{\text{SC}}_{\text{cut}}$ | |
|---|---|---|---|---|---|---|---|---|---|---|---|---|---|---|
| | C | T | C | T | C | T | C | T | C | T | C | T | C | T |
| FO-COUNTERS (20) | 4 | **0.10** | 4 | 2.20 | 4 | 0.11 | 4 | 0.21 | **5** | 3.89 | 3 | 2.37 | 3 | 3.15 |
| FO-COUNTERS-INV (20) | 3 | **0.03** | 3 | 0.66 | 3 | 0.05 | 3 | 0.14 | **4** | 2.23 | 2 | 0.93 | 2 | 0.38 |
| FO-COUNTERS-RND (60) | 14 | 5.82 | 13 | 133.88 | 14 | **0.60** | 14 | 9.79 | **18** | 3.86 | 11 | 2.03 | 13 | 1.90 |
| FO-FARMLAND (50) | 14 | **0.03** | 12 | 1.06 | **16** | 0.06 | 14 | 0.09 | 2 | 147.45 | 4 | 8.74 | 4 | 9.00 |
| FO-SAILING (20) | **2** | - | 1 | - | **2** | - | **2** | - | 1 | - | 1 | - | 0 | - |
| TOTAL (170) | 37 | - | 33 | - | **39** | - | 37 | - | 30 | - | 21 | - | 22 | - |

Table 5: Results on LTs. Coverage ('C') and the time ('T') in seconds are shown. 'T' is averaged over instances solved by all methods.

and Raphael 1968) with $h^C_{\text{IP}}$ and $h^{\text{LM-cut},SEQ}_{\text{LP}}$, the operator-counting heuristic with the LM-cut and state-equation constraints (Kuroiwa et al. 2021). These are the state-of-the-art

admissible heuristics in SCTs. In LTs, we evaluate A* search with the blind heuristic which returns zero if a state is a goal state and $\min_{a \in \mathcal{A}} \text{cost}(a)$ otherwise, $h^C_{\text{IP}}$ ignoring nu-

meric conditions, the interval-based max heuristic ($h^{\text{iimax}}$) (Aldinger and Nebel 2017), and the repetition-based max heuristic ($h^{\text{irmax}}$) (Aldinger and Nebel 2017). To our knowledge, these are the only existing admissible heuristics in LTs. In addition, we evaluate OMTPlan (Leofante et al. 2020), a planner for optimal numeric planning based on optimization modulo theories in LTs. The results in SCTs and LTs are shown in Table 4 and 5, respectively. In both classes of numeric planning tasks, the heuristic search algorithms clearly outperform the MILP based methods. In LTs, OMT-Plan solves more instances than the MILP based methods in domains other than FO-FARMLAND.

## Conclusion

In this paper, we propose a novel MILP compilation of numeric planning using a branch-and-cut algorithm. In the experimental evaluation, the new MILP based method finds the optimal solution faster than the existing MILP based method in some domains and solves more instances in one domain of linear numeric planning tasks. However, the number of solved instances is not increased in any domain of numeric planning tasks with simple conditions. Compared with other approaches, the MILP based methods do not solve more instances than A* with admissible heuristics in any domain.

Developing more competitive MILP compilations is future work. Nevertheless, the performance may be improved without effort in future since the MILP based methods can benefit from the progress on the off-the-shelf solvers. In addition, better admissible heuristics will improve the performance of the iterative time horizon allocation method used in the MILP based methods by providing stronger lower bounds of the time horizon. Pursuing better MILP compilations of planning tasks has an important meaning to the planning community even if they are not competitive with other approaches for now.

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
