# OpenReview forum: "A Branch-and-Cut Approach for a Mixed Integer Linear Programming Compilation of Optimal Numeric Planning"
_icaps-conference.org/ICAPS/2021/Workshop/HSDIP — HSDIP 2021_

### Official Review · AnonReviewer1 · 2021-05-25

**Confidence:** 4
**Overall Score:** Strong Accept

**Review:**

### Summary

The paper extends a branch-and-cut procedure for MIP-based STRIPS planning to MIP-based numeric planning. The relaxing assumption in the MIP is that multiple actions can be executed in parallel in a single step. If a solution is found where the actions used in a step are not serializable, because they have a cyclic dependency, a constraint is added to break this cycle. This way, adding an exponential number of cycle-breaking constraints in the initial model can be avoided and a smaller time horizon is sufficient as actions can be executed in parallel. The experiments demonstrate that often cycles can be completely
avoided (by precomputing and removing cycles of length 2) and in the cases where there are cycles, only a few constraints have to be added. While the performance of the resulting method is not a huge advantage over the method without branch-and-cut, there are cases where a benefit is visible.

The theoretical contribution of the paper is interesting and I spotted no formal mistakes. I did not understand some parts (see below) but those where mostly based on previous work that was already reviewed, so I don't suspect an error here.

The paper is well written but very technical with a lot of content to go through and a lot of notation to keep in mind while reading. I don't have a good suggestion of how to improve this but anything you can do to simplify the paper would be appreciated.

In summary, this is a high-quality paper with an interesting theoretical contribution that should clearly be accepted at HSDIP.


### Technical Issues

For the MILP model for numeric planning, I wasn't able to see the correspondence between plans an MILP solutions. I can see that every plan can be transformed into a solution of the MILP but I'm less clear on whether any MILP solution can be transformed into a plan. The reason why I'm unsure is the lower and upper bounds for the numeric variable values. It seems as though the bounds could be weaker in later time steps and thus allow solutions that do not correspond to plans. This may be explained in the paper this is based on (I didn't look into it). I have the same doubts on the updated constraints that consider the effects of parallel actions.


### Experiments

I was somewhat surprised to see the benchmarks restricted to unit-cost domains as the theoretical part specifically mentions 0-cost actions and discusses how to deal with them.


### Minor comments

* In the definition of s[a]_p, the usual convention is to have add effects win over delete effects (add-after-delete semantics), so the order of set difference and union should be changed (adding parentheses would help as well). Of course it doesn't matter if add(a) and del(a) are disjoint but I'd still write the definition in this way to avoid confusion.
* Missing space in "planas" in the first paragraph of the background.
* The definition of "mutex(a)" is not sufficient to cover the conditions mentioned before equations (3)-(4). It doesn't consider numerical conditions and only considers applying b after a. In my opinion, the term "mutex" should not be used for such an asymmetric definition. I'd rather call such actions, the actions that a interferes with. Apart from the name, the sentence before (3) should be weakened (the constraint does not ensure the condition).
* The text mentions that constraints (12)-(16) enforce preconditions and effects. However, in all these constraints, variables y (talking about facts) occur with an index t+1 while the variables x (talking about actions) occur with an index t. The constraints thus all talk about applying an action at time t and the values of facts after applying the action. I see no way how a precondition of an action would propagate information to variables y with a lower index, i.e., how these constraints ensure that the precondition of a selected action are satisfied in the state where it is applied. This happens with a somewhat indirect interaction through constraint (19).
* In the first sentence of "Branch-and-Cut for STRIPS", I'd move the reference to van den Briel et al. after "a branch-and-cut framework for classical planning". In the current phrasing, it is easy to read the sentence in a way where the references are assigned to the wrong statements.
* When discussing constraint (27), a forward reference to the section "Big-M Values" would be helpful.

---

> ### Author Response · Authors · 2021-06-01
> **Response to Reviewer1**
>
> Thank you for the review.
>
> >In the definition of s[a]_p, the usual convention is to have add effects win over delete effects
>
> Thank you for the suggestion. We will use the standard notation.
>
> >For the MILP model for numeric planning, I wasn't able to see the correspondence between plans and MILP solutions.
>
> Piacentini et al. (2018a) proved that every feasible solution for the MILP model corresponds to a plan in Proposition 4.2. in the appendix (https://tidel.mie.utoronto.ca/pubs/TechReport2018A.pdf).
> In particular to your concern, we think there is no problem; as you stated, $m_{v,t}$ and $M_{v,t}$, the lower and upper bounds for numeric variable values at time step t, could be weaker in later time steps. However, these bounds are used in Big-M constraints in Constraints (28)-(31) to cancel constraints related to unapplied actions. When no action with linear effects is applied, RHS of Constraints (28) and (29) are the same since big-M values (the last terms) are cancelled, and RHS of Constraints (30) and (31) impose nothing. When an action with linear effects is applied, RHS of Constraint (28) and (29) impose nothing, and RHS of Constraints (30) and (31) are the same. Therefore, the changes of numeric variable values are equal to the numeric effects.
>
> >I was somewhat surprised to see the benchmarks restricted to unit-cost domains as the theoretical part specifically mentions 0-cost actions and discusses how to deal with them.
>
> As we mentioned in the paper, the MILP models do not guarantee the optimality in problems with 0-cost actions (“Note that this method is not applicable in planning tasks with $\min_{a \in \mathcal{A}}$ cost(a) = 0.” on p.5) due to the iterative time horizon allocation method. Therefore, we use unit-cost versions of the domains.
>
> >The definition of "mutex(a)" is not sufficient to cover the conditions mentioned before equations (3)-(4).
>
> We admit that this naming and description are a bit confusing. As you pointed out, $b \in \mathsf{mutex}(a)$ if applying $b$ after $a$ is impossible. However, the constraint ensures the condition; since $a$ and $b$ are not applicable at the same time step if $a \in \mathsf{mutex}(b)$ or $b \in \mathsf{mutex}(a)$, they are applicable only if $a$ can be applied after $b$ and $b$ can be applied after $a$.
>
> >I see no way how a precondition of an action would propagate information to variables y with a lower index, i.e., how these constraints ensure that the preconditions of a selected action are satisfied in the state where it is applied. This happens with a somewhat indirect interaction through constraint (19).
>
> As written in p.3 ($y_{p,t}^{pa}$ indicates that $p$ is required as a precondition and not deleted from the state by actions applied at $t-1$),  $y_{p,t+1}^{pa} = 1$  indicates that p exists at time step $t$ and $t+1$. If $x_{a,t} = 1$, Constraint (15), $x_{a,t} <= y^{pa}_{p,t+1}$, requires that precondition p of a is satisfied at t and not deleted.
>
> >In the first sentence of "Branch-and-Cut for STRIPS", I'd move the reference to van den Briel et al. after "a branch-and-cut framework for classical planning".
> >When discussing constraint (27), a forward reference to the section "Big-M Values" would be helpful.
>
> We will revise the texts accordingly.

---

> > ### Comment · AnonReviewer1 · 2021-06-02
> > **Thank you for the clarifications**
> >
> > Thank you for your response.
> >
> > I was convinced on the technical aspects and see no more technical issues.
> >
> > I had misread the discussion of expanding the horizon and thought it would be a solution to 0-cost action but I see now that it isn't, so it makes sense to consider non-zero action costs. However, instead of unit-costs, you could have used the "plus one" model that LAMA uses (if there are no zero-cost actions, use the original cost function, otherwise, add 1 to all costs). This way, you could evaluate the way your method deals with more general action costs.
> >
> > Regarding the definition of mutex, I suggest to change this to a relation, where "(a,b) not in mutex iff both orders of executing a and b are possible and reach the same state". Constraint (3) would then be quantified over all (a,b) in mutex and definition (4) would have to be updated to cover both directions. I agree that the current version is correct but it is very confusing. If you want to stick to the current definition, I'd at least consider renaming "mutex" to "interfering".
> >
> > Finally, regarding constraints (12)-(16), I also agree that they are correct. The issue I had was just with the formulation "Constraints  (12)-(16)  enforce  preconditions  and  effects  of actions". This is technically true but misleading, I think, because without constraint (19) there is no connection between the time steps. So without constraint (19) it would be possible to select actions in earlier time steps that violate preconditions. In my opinion the sentence should be reworded but I don't have a good suggestion. Maybe discussing the constraints in more detail would help to make it more clear.
> >
> > One aspect that makes understanding the model complicated and that is unusual compared to other models is that variables y talk about two time steps in the plan execution, so the time steps in the model (y_t, y_{t+1}) are different than the time steps in the plan execution (e.g., y_3 talks about variables before and after the third action execution). Discussing this might also help the reader.

---

> > > ### Author Response · Authors · 2021-06-02
> > > **Response to the Comment**
> > >
> > > Thank you for the detailed description.
> > > We will revise the definition and clarify the explanations.

---

### Official Review · AnonReviewer2 · 2021-05-26
**Thorough treatment; on point; a bit crammed**

**Confidence:** 4
**Overall Score:** Weak Accept

**Review:**

The authors introduce a MILP model for a restricted class of numeric planning, which primarily focuses on being able to reduce the number of required layers in the temporal encoding. They do this by shifting from a forall semantics on simultaneous actions to the exists semantics, and further introduce the element of branch-and-cut to remove cycles found in the causal support at each layer.

The work is relevant to HSDIP, and the model seems to be complete. However, in this format, it is extremely dense. There isn't a lot of space to fully expand on what the parts of the model mean, and it feels as though a journal submission might be the best format for the work.

Ultimately, the results are not that great for the elements introduced. There could be some deeper exploration as to where the problem lies (unsolvable iterations? final iteration?), but the evaluation is otherwise fairly complete and indicative of when this approach works. Seeing only 3 domains pose an issue for cycle removal is an interesting result, even if it leads to little improvement across the full benchmark set.

On the topic of benchmarks, it would be interesting to see if there is a synthetic domain that exemplifies the strength of what's being proposed. Over both the previous MILP model and (ideally) the heuristic-based approaches.

I'm leaning towards accepting, and below are some minor points to fix / questions for the authors.

- I understand that it complicates matters having the ADD and DEL sets overlap, but it's probably worth using the standard STRIPS state update: (s_p \ del(a)) U add(a)

- Table 2 is really nice to have! A general list of the differences between the approaches would be good too (rather than having to look up the subtle differences between equation #'s')

- For many of the domains, it seems as though both approaches achieve full coverage. Is there no way to sc

- You state"better admissible heuristics will improve the performance...by providing stronger lower bounds of the time horizon". What's the ceiling on this? I.e., if you had an oracle give you the /precise/ bound, how difficult would it be to solve that problem, compared to the other heuristic-based methods?

- The proposed encoding seems strikingly similar to exists-step plans ( https://link.springer.com/chapter/10.1007/978-3-540-76928-6_26 ). I realize that you are in the numeric setting, but I highly recommend including a comparison.

---

> ### Author Response · Authors · 2021-06-01
> **Response to Reviewer2**
>
> Thanks for your helpful comments.
>
> >I understand that it complicates matters having the ADD and DEL sets overlap, but it's probably worth using the standard STRIPS state update: (s_p \ del(a)) U add(a)
>
> We will use the standard notation.
>
> >Table 2 is really nice to have! A general list of the differences between the approaches would be good too (rather than having to look up the subtle differences between equation #'s')
>
> We will try to include the list if there is sufficient room.
>
> >For many of the domains, it seems as though both approaches achieve full coverage. Is there no way to sc
>
> It seems that the comment is cut-off for some reason. Could you reexplain the question?
>
> >You state"better admissible heuristics will improve the performance...by providing stronger lower bounds of the time horizon". What's the ceiling on this? I.e., if you had an oracle give you the /precise/ bound, how difficult would it be to solve that problem, compared to the other heuristic-based methods?
>
> This is an interesting question. If we have a precise upper bound, we can always guarantee the optimality even with 0-cost actions and do not need to solve multiple MILP models with different time horizons. However, if the bound is too large, the MILP models use an unnecessarily large number of variables and constraints, which can lead to performance degradation. Therefore, we suspect that there is a trade-off between using lower and upper bounds.
>
> >The proposed encoding seems strikingly similar to exists-step plans (https://link.springer.com/chapter/10.1007/978-3-540-76928-6_26 ). I realize that you are in the numeric setting, but I highly recommend including a comparison.
>
> We will cite the paper. The difference between exists-step plans and the branch-and-cut model for classical planning that we are based on is explained in Section 5 (Related Work) of that paper.

---

> > ### Comment · AnonReviewer2 · 2021-06-01
> > **Cutoff completion & other stuff**
> >
> > For many of the domains, it seems as though both approaches achieve full coverage. Is there no way to sc**ale up the problem set difficulty so that you can see where one approach falls off in coverage while the other continues.**
> >
> > > we suspect that there is a trade-off between using lower and upper bounds.
> >
> > You can test this directly, at least for those problems where an optimal solution is found -- just one further MILP call with precise (lower==upper) bounds to see what the difficulty would be. I can even imagine scaling the upper bound between the optimal and computed, just to see the trend in difficulty it creates.

---

> > > ### Author Response · Authors · 2021-06-01
> > > **Response to the Comment**
> > >
> > > >For many of the domains, it seems as though both approaches achieve full coverage. Is there no way to scale up the problem set difficulty so that you can see where one approach falls off in coverage while the other continues
> > >
> > > Gardening-SAT and Sailing-SAT are scaled-up versions of Gardening and Sailing. For Farmland, there also exists Farmland-SAT but no instance is solved by our planners due to out of memory during grounding.
> > > In Counters and Farmland, since the two MILP models happen to use the same set of constraints (the numbers of constraints are exactly the same in Table 3), there will be no difference if we use scaled-up versions.
> > > In Gardening-SAT and Sailing-SAT, the two methods are the same in coverage, and sets of solved instances are also the same. We may observe differences in coverage by using instances that are more difficult than the solved but easier than the unsolved.
> > > As you suggested in the review, it is worth trying to generate appropriate problems to contrast differences.
> > >
> > > >You can test this directly, at least for those problems where an optimal solution is found -- just one further MILP call with precise (lower==upper) bounds to see what the difficulty would be. I can even imagine scaling the upper bound between the optimal and computed, just to see the trend in difficulty it creates.
> > >
> > > Thank you for the practical suggestion.

---

> > ### Comment · AnonReviewer1 · 2021-06-02
> > **exists-step plans**
> >
> > I want to echo my co-reviewer's point that a comparison with exists-step plans would be useful. I was also thinking about this while reading but forgot to add it to my review. Looking at the paper by Wehrle and Rintanen, they do not discuss the similarities and differences in that much detail. I would be interested in reading more about this: in what ways are the two methods the same and are there differences apart from the representation of the constraints?

---

> > > ### Author Response · Authors · 2021-06-02
> > > **Similarities and differences between the branch-and-cut model and the eixsts-step plans encoding**
> > >
> > > We think that the action precedence graph in the branch-and-cut model and the disabling-enabling graph in the exists-step encoding are very similar. However, while the former considering the reachability of states, the latter is not, so this may create differences. As discussed by Wehrle and Rintanen, another difference is that the branch-and-cut model prohibits simultaneous applications of actions that form a cycle while the exists-step plans encoding allows by imposing a fixed ordering on actions in all strongly connected components of the graph. Using a fixed ordering in the branch-and-cut model can be future work.

---

> > > > ### Comment · AnonReviewer1 · 2021-06-03
> > > > **Response to comment**
> > > >
> > > > Thanks. I think this would also be a good discussion to add to the paper.

---

### Decision · Program_Chairs · 2021-06-10

**Decision:**

Accept

**Comment:**

The reviewers agreed that this submission fits well into the HSDIP program, so we are happy to accept it. The main point of criticism in both reviews was that the paper is very dense and we encourage the authors to follow suggestions in the reviews about improving this.